# How Bronchoscopic Dye Marking Can Help Minimally Invasive Lung Surgery

**DOI:** 10.3390/jcm11113246

**Published:** 2022-06-06

**Authors:** Matthieu Sarsam, Jean-Marc Baste, Luc Thiberville, Mathieu Salaun, Samy Lachkar

**Affiliations:** 1Department of General and Thoracic Surgery, CHU Rouen, F-76000 Rouen, France; matthieu.sarsam@chu-rouen.fr (M.S.); jean-marc.baste@chu-rouen.fr (J.-M.B.); 2Normandie University, University of Medicine and Pharmacy of Rouen, UNIROUEN, INSERM U1096, FHU REMOD-VHF, F-76183 Rouen, France; 3Department of Pulmonology, CHU Rouen, F-76000 Rouen, France; luc.thiberville@univ-rouen.fr (L.T.); mathieu.salaun@univ-rouen.fr (M.S.); 4QuantIF Team, LITIS Lab EA4108, UNIROUEN, Normandie University, F-76000 Rouen, France

**Keywords:** solitary pulmonary nodule, bronchoscopy, minimally invasive surgery, indocyanine green, methylene blue, radial EBUS, VAL-MAP, EMB

## Abstract

In the era of increasing availability of high-resolution chest computed tomography, the diagnosis and management of solitary pulmonary nodules (SPNs) has become a common challenging clinical problem. Meanwhile, surgical techniques have improved, and minimally invasive approaches such as robot- and video-assisted surgery are becoming standard, rendering the palpation of such lesions more difficult, not to mention pure ground-glass opacities, which cannot be felt even in open surgery. In this article, we explore the role of bronchoscopy in helping surgeons achieve successful minimally invasive resections in such cases.

## 1. Surgery in Lung Cancer

The prevalence of solitary pulmonary nodules (SPNs) can be as high as 50% in lung screening programs [1], and their successful management can decrease lung-related mortality by 20%, as shown by The National Lung Screening Trial [2].

These results are confirmed by the NELSON study, where 15,792 participants were randomly assigned to either periodic low-dose CT screening or no screening. After 10 years of follow-up, lung cancer mortality was lower in the screening group than in the control group, both among men (lower by 24%) and among women (lower by 33%) [3]. These numbers underline the importance of early diagnosis and treatment.

While lobectomy is still the gold-standard treatment for early-stage non-small-cell lung carcinoma (NSCLC) in patients with normal pulmonary function tests, segmentectomy has become an established choice of treatment in those with limited functioning, advanced age, and important comorbidities. Its role can also be diagnostic, notably in the case of infra-centimetric lesions, ground-glass opacities (GGOs), or lesions located deeply within the lung parenchyma.

Literature review and metanalysis show that survival is comparable whether the patient is treated by segmentectomy or lobectomy, for early-stage lung cancer [4].

Wedge resections can also be considered to treat lung cancer in patients with limited functioning or lung metastases, or to diagnose a lesion of an unknown histopathology [5,6,7].

The key to a successful infra-lobar resection lies in preoperative planning, the correct localization of the lesion, and securing good margins around it [8,9].

In a multicenter study by Sato et al. published in 2018, the authors highlighted the importance of this concept, and that the depth of the required margin was the most significant factor associated with resection failure in a series of 203 lesions requiring sublobar resections [10].

While traditional surgical teaching emphasizes the importance of bimanual palpation of such lesions, minimally invasive techniques have become more and more popular, and do not allow for palpation.

For example, in France, the number of segmentectomies performed via minimally invasive surgery, compared to open thoracotomy, has rapidly increased over the past 10 years (Figure 1).

To overcome these difficulties, various methods of SPN localization have been developed, including bronchoscopy-based techniques, or percutaneous ones, such as CT-guided radiotracer injection [12], CT-guided wire localization [13], CT-guided fiducial placement and fluoroscopic localization [14], ultrasonography [15], and percutaneous methylene blue (MB) injection [16].

Practically speaking, CT-guided procedures are cumbersome, and require a high degree of coordination between surgeons, anesthetists, and radiologists, as the technical platform of CT-guided procedures is often located outside the operative theater.

Furthermore, complications such as pneumo- or hemothorax can occur. In a series of 174 CT-guided lipiodol markings reported by Watanabe et al., 16 patients suffered from pain requiring analgesia, 11 from bloody sputum, and 30 from pneumothorax including 11 requiring chest drainage; additionally, 1 patient required an emergency surgery for hemopneumothorax [17]**.**

In this review, we explore the different bronchoscopy-based methods, as well as their respective efficacies, which have been the subject of many publications; of these, we cite a recent metanalysis by Yanagia et al. in 2020 [18].

## 2. A Word on Endobronchially Delivered Markers

Endobronchially delivered markers range from dye agents (indigo carmine (IC), Methylene blue (MB), and indocyanine green (ICG)), to the placement of coils and fiducials [14,19,20].

The choice of dye agent depends on many factors, including the time between the localization procedure and surgery, as some agents such as indigo carmine last 2–3 days [21], while others such as methylene blue can diffuse to adjacent tissues rapidly [22,23].

The use of ICG, on the other hand, requires a near-infra-red (NIR) camera. It has been reported as an intrabronchial tattooing agent [24]. Moreover, it can be used intravenously, either intraoperatively to delaminate the intersegmental plane [25], or 24 h prior to surgery to detect nodules as small as 0.2 cm and as deep as 1.3 cm from the pleural surface [26].

## 3. Electromagnetic Navigation Bronchoscopy

Electromagnetic navigation bronchoscopy (ENB) is often compared to a global positioning system (GPS) for the lung. This allows the physician to navigate through the bronchi, locate biopsy sites, and ultimately treat endobronchial distal lesions. The system uses a bronchial map and 3D virtual navigation based on a CT scan.

The technique first requires a planning phase, where the two-dimensional slices of the CT scan are converted into 3D models, allowing the physician to mark the lesion and to obtain a “road map” through the bronchial tree to the lesion.

A field generator is placed behind the patient, who is lying in the dorsal decubitus position on the operative table, and emits a low-strength electromagnetic field that passes through the patient’s body. When a sensor is introduced into the field, the current is generated within its coils, allowing for its precise localization (Figure 2).

The planning phase depends mainly on the quality of the CT scan, with the best results being obtained from infra-millimetric slices, as these better show the small bronchi and their bifurcation, and thus generate more precise “road maps”.

ENB navigation was first reported in 2006. In their study, Schwarz et al. demonstrated the feasibility and safety of the procedure, as well as a diagnostic yield of 69% [27]. There were no procedure-related adverse events.

Recently, the NAVIGATE study, a prospective multicenter study, evaluated ENB in 1388 consecutive subjects, enrolled at 37 sites in the USA and Europe, for biopsy (1329; 95.7%), fiducial marker placement (272; 19.6%), dye marking (23; 1.7%), or lymph node biopsy (36; 2.6%) [28]**.**

In terms of complications, the ENB-related pneumothorax rate was 4.9%, and bronchopulmonary hemorrhage and respiratory failure rates were 1.0 and 0.6%, respectively [29,30].

Bowling et al. performed a sub-group analysis of the dye-marking group in the USA cohort of the NAVIGATE study [31]. Among the patients, 23 subjects (24 lesions) underwent dye marking as a preparation for surgical resection. The median nodule size was 10 mm. Most lesions (95.5%) were located in the peripheral third of the lung, at a median distance of 3.0 mm from the pleura. Dye marking was adequate for surgical resection in 91.3% of lesions. Surgical biopsies were malignant in 75% (18/24) of lesions.

Despite good results, this technique has some limitations. First, localization is not carried out in real time, but rather upon CT scan reconstruction, and, consequently, is dependent on its quality. This was studied by Chen et al. [13], who looked at 85 lesions in 46 different patients in full inspiratory vs. full expiratory CT scans. They showed an averaged motion of 17.6 mm of all pulmonary nodules, which was more prominent in lower lobe lesions due to diaphragmatic movement.

The second limit relates to the use of a therapeutic bronchoscope, which results in less selective catheterization of the subsegmental bronchi, and finally the specific OR preparation and expensive disposable material required by ENB.

## 4. Virtual-Assisted Lung Mapping (VAL-MAP)

Sato et al. reported the use of virtual bronchoscopy and a bronchoscopic multispot dye-marking technique using three-dimensional virtual imaging, for precise thoracoscopic sublobar lung resection with safe surgical margins [32].

Three-dimensional reconstruction of computed tomography (CT) data is first performed to generate virtual bronchoscopy images, as this helps the operator design multiple locations for dye marking on the lung surface and define their different routes.

The procedure is achieved under local anesthesia and sedation. Once the bronchoscope reaches the target bronchus, using the predefined virtual bronchoscopy route, a metal-tip catheter is inserted into the bronchus and advanced to the pleura, under fluoroscopic guidance, followed by injection of 1 mL of indigo carmine. This is repeated to complete all the planned markings. The procedure is usually performed within 48 h before surgery, and is followed by a CT scan and 3D reconstructions to confirm the localization of markings within 2 h after the mapping procedure.

In the original article published in 2014, out of the 95 marking attempts for 37 tumors in 30 patients, 88 (92.6%) were identified and contributed to the surgery, with a successful resection rate of 100%. No clinical complications were reported.

In clinical practice, dye marking is not always successful; sometimes, the planned location is not correctly marked and/or not visible on the lung surface. Hence, the role of the post-mapping CT scan in conventional VAL-MAP is significant. Indeed, Sato et al. analyzed 43 markings in 11 patients and found that the average difference between the predicted and actual marking locations was 30 mm. Despite this discrepancy, all lesions were successfully removed thanks to the use of post-mapping CT scan guidance, as it gave the surgeons a 3D understanding of the actual location of the marks and their distances from the lesion, rather than their planned virtual locations [33].

This technique also presents logistical challenges regarding the post-dye-marking CT scan. The authors have reported a simpler modified method that uses ENB in order to avoid post-mapping CT scans [34]. All procedures were performed under general anesthesia in the operative room. Two groups were compared: The first was the non-adjustment group, where surgery was performed immediately after ENB dye marking, while in the second group, the locational information was transferred to a radiology workstation to construct an adjusted 3D image.

The accuracy was graded intraoperatively.

The authors conclude that ENB VAL-MAP quality was improved by adding on-site adjustment, i.e., without post-mapping CT scans, achieving clinical outcomes similar to conventional VAL-MAP.

One of this technique’s limitations (as for ENB) is that the 2D tattooing on the lung surface jeopardizes acceptable resection margins, as the lesion is located deeper into the lung parenchyma. In order to overcome this, VAL-MAP 2.0 was developed. With VAL-MAP 2.0, the difference lies in placing one or two microcoils (visible on intraoperative fluoroscopy) in the most peripheral bronchus, leading to the tumor, thus securing better margins [35].

The first clinical trial to investigate the effectiveness and safety of VAL-MAP 2.0 is currently underway, and it is hoped that it will enable more accurate resectioning with sufficient margins, even of deeply located lesions [36]. The study, however, is a single-arm study, containing biases in patient selection and outcome comparison, compared to other techniques.

While short-term results have been repeatedly reported, Yamaguchi et al. recently published a study focusing on the long term. The authors retrieved clinical data pertaining to the 264 patients (sublobar resection after VAL-MAP) eligible of the 663 ones enrolled in two prospective short-term studies. The 5-year local recurrence-free rate was 98.4%, and the 5-year overall survival (OS) rate was 94.5% [37].

To date, VAL-MAP appears to be the most precise technique for preoperative localization of small peripheral lung nodules.

However, it cannot be performed immediately before the surgery or during the same surgical procedure, leading to increased time and resources for each procedure. Therefore, even if it is highly efficient, this technique appears difficult to implement, at least in the near future, in most surgical centers, which conflicts with the growing issue of peripheral lung nodules that need to be removed.

## 5. Virtual Bronchoscopy Combined with Radial EBUS

Radial EBUS combined with virtual bronchoscopy can easily be used for pleural dye marking, as it is easy to perform in the operative room, relatively cheaper than other techniques, and reproducible, especially for wedge resections or segmentectomies.

The procedure starts with uploading the CT scan into a virtual bronchoscopy program, allowing the pulmonologist, after careful study of the frontal, coronal, and sagittal sections, to mark the target. The software then reconstructs a pathway leading to the lesion. This technique does not provide real-time navigation, and the pulmonologist memorizes the pathway and can consult it on the computer at all times in case of doubt; it is a peri-operative form of assistance. Note that this is the main difference between virtual navigation and augmented navigation (i.e., EMN).

Then, the bronchoscopist follows the predefined route, and reaches the most distal bronchus. The guide sheath with the r-EBUS probe is then inserted into the working channel and pushed towards the lesion in order to reach the subpleural space. The probe is then removed and 1 mL of methylene blue (5 mg/1 mL) is injected and rinsed with 20 mL of air. When the NIR camera is available in the operative room, double dye marking using 0.5 ml of MB and 0.5 of IGG can be used, giving the surgeon full visibility on the lesion at all times, during normal as well as infra-red vision (Figure 3).

In our institutional experience, this method adds no more than 10 min of procedure time to the surgery in experienced hands. In fact, between April 2016 and June 2017, all anticipated difficult minimally invasive sublobar resections of peripheral lesions (*n* = 25) were marked using this method.

The dye was visible on the pleural surface in 24 cases. Histological diagnosis and free margin resection were obtained in all cases [38].

Note that the purpose of the procedure is to mark the nodule’s area rather than the nodule itself (which is far from the pleura in the majority of cases), thus helping the surgeon to be more confident regarding the nodule localization.

## 6. Robot-Assisted Broncho-Navigation

There are currently two platforms available on the market, MONARCH™ by Auris and ION^TM^ by Intuitive. Both use a combination of direct visualization from the internal probe’s eye in addition to an endobronchial ultrasound and navigational guidance. Robotics can help achieve a more stable navigation and give an enhanced maneuverability, especially in difficult and angulated small bronchi.

The setting starts with a planning phase where conventional CT scans are converted to 3D modules, in which the interventional bronchoscopist can locate the lesion. The software then creates a navigation map that can be uploaded onto the robotic platform. One of the main advantages is the ability to have real-time navigation throughout the procedure, even during the biopsy.

These recent technologies provide more access to peripheral lung lesions, as shown by the REACH study [39], which aimed to assess the reaching rate of a robotic endoscopic system compared with conventional thin bronchoscopes of the same diameter within human cadaveric lungs. The robot-assisted bronchoscope could access all 18 segments of the lung and reached an average of 4.2 cm further than the conventional one.

The real question remains, which is whether this further navigation can help enhance our diagnostic abilities of these nodules. The current clinical data are limited, but the results of the PRECISION-1 STUDY [40] should be highlighted. This prospective single-blinded randomized controlled comparative trial aimed to compare an ultrathin bronchoscope (3.0 mm) with r-EBUS (Olympus), electromagnetic navigation (Superdimension, Medtronic) with a 6.0 mm bronchoscope, and robotic bronchoscopy with a 3.5 mm (Ion Endoluminal System, Intuitive) bronchoscope in a human cadaver model of guided bronchoscopy for the diagnosis of peripheral pulmonary nodules. The procedures were performed by six experts in advanced diagnostic bronchoscopy, who used the three modalities to biopsy 20 lung nodules (60 procedures in total).

The needle puncture of the nodules was confirmed by visualizing the needle in the lesion on cone-beam CT. After needle deployment, the bronchoscopist was blinded to cone beam CT results.

The main results show that nodule puncture rate by robotic bronchoscopy (80%) was significantly higher compared with both EMN (45%, *p*-value 0.022) and ultrathin bronchoscopy with EBUS (20%, *p*-value < 0.001). Nodule localization was successful in 65% of ultrathin bronchoscopy with EBUS, 85% of EMN, and 100% of robotic bronchoscopy cases.

As with RATS, cost is the main limiting factor that will hinder the adoption of this new technology. Logistics issues must also be considered as these platforms are immobile, rendering immediate pre-operative dye marking rather difficult if the patient is to be operated on in a different operative room.

A summary of these methods is provided in Table 1.

## 7. Conclusions

Bronchoscopic dye marking represents a valuable and safe solution to localize small lung nodules in the setting of minimally invasive surgery.

The authors believe that it represents one of the cornerstones of modern surgical management, alongside 3D reconstructions and virtual reality.

The best technique should be precise, time-efficient, and easily reproducible in one’s own center. The future seems promising with the development of robot-assisted techniques and new dyes.

## Figures and Tables

**Figure 1 jcm-11-03246-f001:**
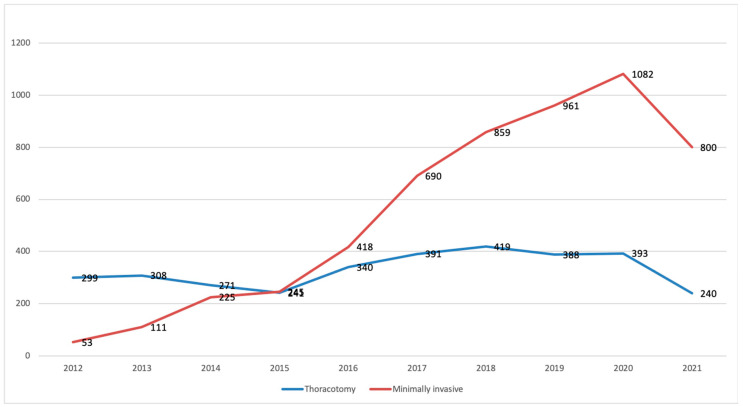
Number of segmentectomies based on a surgical approach in France between 2009 and September 2021. Data extracted from Epithor database [11], in red: minimally invasive surgeries.

**Figure 2 jcm-11-03246-f002:**
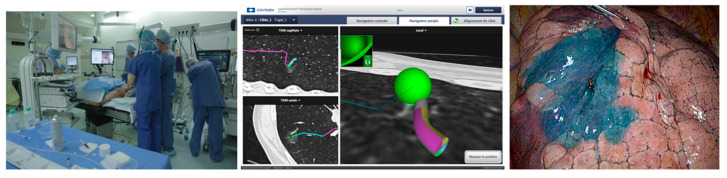
Dye marking using ENB, courtesy of Dr. Agathe Seguin-Givelet.

**Figure 3 jcm-11-03246-f003:**
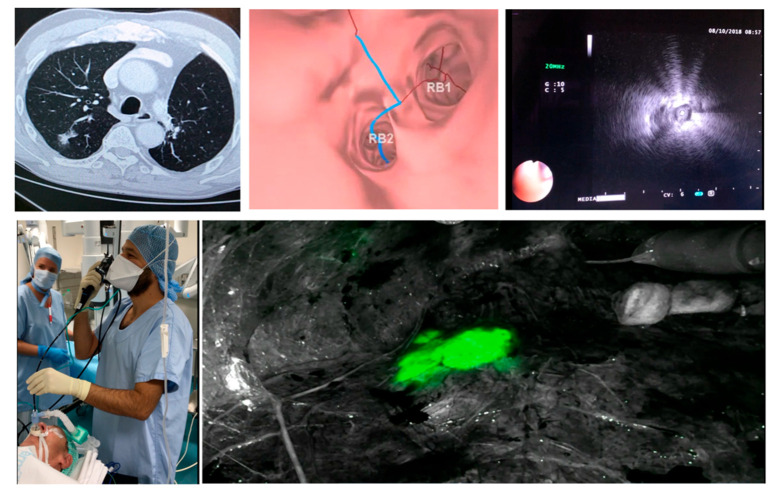
Dye marking using virtual bronchoscopy + R EBUS, courtesy of Dr. Samy Lachkar.

**Table 1 jcm-11-03246-t001:** Summary of bronchoscopic techniques in nodule localization.

	Dye Marking Accuracy	Requirements	In Clinical Practice	Limitations	Cost
**Virtual bronchoscopy combined with Radial-EBUS**	(85-96%) depending on the publication.	Chest CT scan (preferably infra millimetric slices). EBUS catheters and virtual bronchoscopy software.	Nodule dye marking can be done at the OR just before surgery	Precision	€ *
**Electromagnetic Navigation Bronchoscopy (ENB)**	91.3% according to the Navigate study	Chest CT scan (preferably infra millimetric slices) ENB navigation system and planning software	Nodule dye marking can be done at the OR just before surgery Requires placing a field generator behind the patient	Cost CT to body divergence	€€ *
**Virtual assisted lung mapping (VAL-MAP)**	92.6 %(Currently considered to be the most precise method).	Pre interventional CT scan Virtual planning software post interventional CT scan In VALMAP 2.0 ENB navigation system	Dye marking is usually done within 48 h of the surgery.Control by CT scan is required before surgery	Cost Logistics (2 CT scans). Inability to mark the lesion intraoperatively	€€ *
**Robot-assisted broncho-navigation**	Cadaveric small studies show 100% accuracy rate More clinical data are needed to validate this aspect.	- Robotic platform - Special infrastructure at the OR. - Add cone beam CT scan at the OR.	Allows intraoperative nodule marking and biopsies in big OR (To fit surgical robot, bronchoscopy robot and Cone beam CT).	Cost Nodule dye marking and surgery would probably be performed in 2 different OR as it’s hard to fit both robots in one room.	€€€ *

€ * is less expensive, €€ * more expensive, €€€ * most expensive.

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
