# Peer review of "How Bronchoscopic Dye Marking Can Help Minimally Invasive Lung Surgery"

_jcm, 2022, doi:10.3390/jcm11113246_

Round 1

Reviewer 1 Report

Sarsam et al. provide a nice review of the current modalities available for localization of lung nodules in the era of more minimally invasive surgical techniques.

One area that should be mentioned is the cost of the various modalities and the learning curve for for performing these. Likewise, some summary of the accessibility to these techniques among institutions would also be recommended.

An additional comment would be how these modalities replace or compliment the intraoperative assessment of diagnostic tissue via pathologic review.

An additional suggestion would be to create a table summarizing these different modalities. As the paper reads, it is hard to keep all these organized once the reader moves on to the next topic. A table would be a nice way to have this information presented in a localized fashion.

Author Response

We thank the reviewer for his kind comments.

One area that should be mentioned is the cost of the various modalities and the learning curve for performing these. Likewise, some summary of the accessibility to these techniques among institutions would also be recommended.

This is a very interesting point, however, in the literature, we didn't found data to address the issue of learning curve. Regarding the cost, it is very hard to come to prices ranges as they differ from country to another. However, we have added that to a summary table to give a global idea about the issue.

In regards to accessibility, this is institution-dependent. But in our own opinion, the easiest technique to set up is Virtual bronchoscopy r-EBUS.

An additional comment would be how these modalities replace or compliment the intraoperative assessment of diagnostic tissue via pathologic review.

We haven’t discussed this interesting issue as the main aim of our paper was to address nodules localisation.

An additional suggestion would be to create a table summarizing these different modalities. As the paper reads, it is hard to keep all these organized once the reader moves on to the next topic. A table would be a nice way to have this information presented in a localized fashion.

Many thanks for this suggestion; as request, a table has been created and added to the main text.

Reviewer 2 Report

It is a very interesting and good work , with a review of actual bronchoscopic techniques of  dye marking and it is very important to chose the most minimally invasive surgical method. 

the last citation should be with small letter. 

Author Response

Thanks a lot for your kind comment. We are happy that the paper was up to the reviewer’s standards.

the last citation should be with a small letter. 

Thanks a lot for noticing this. The format was changed accordingly.

Reviewer 3 Report

Dear Editor and Authors,

As a practicing thoracic surgeon I am very keen to see concise and well-constructed reviews regarding new technologies and novel solutions to surgical problems. Therefore, I was very interested to read the article by Dr. Matthieu Sarsam and his colleagues from the Department of General and Thoracic Surgery at the University Hospital of Rouen in Rouen, France.

This is indeed a well-researched and well written work presenting an update on the modern bronchoscopic techniques which can be utilized to localize and mark lesions for minimally invasive excision.

In my experience and purely on anecdotal evidence I will agree with the authors that Virtual bronchoscopy combined with Radial-EBUS is probably the most feasible technique to be clinically used both in terms of cost and in terms of logistics. I would have liked to authors to explore a bit more the comparison between the techniques and add the pros and cons/limitations of each one in a more concise and easy to present way (maybe a table or a small discussion piece?).

My kindest regards to all.

Author Response

Dear reviewer,

The authors of the paper thank you for your kind words. Indeed, the aim was to provide a concise and a clear summary of different techniques that can be exploited in surgery.

In order to make things clearer, we have created and add a summary table to the text.

Best regards

Round 2

Reviewer 1 Report

The table looks great. Much easier to follow and a nice source to be able to reference.